# Discovering conflicting groups in signed networks

**Ruo-Chun Tzeng**
KTH Royal Institute of Technology
`rctzeng@kth.se`

**Bruno Ordozgoiti**
Aalto University
`bruno.ordozgoiti@aalto.fi`

**Aristides Gionis**
KTH Royal Institute of Technology
`argioni@kth.se`

## Abstract

Signed networks are graphs where edges are annotated with a positive or negative sign, indicating whether an edge interaction is friendly or antagonistic. Signed networks can be used to study a variety of social phenomena, such as mining polarized discussions in social media, or modeling relations of trust and distrust in online review platforms. In this paper we study the problem of detecting $k$ conflicting groups in a signed network. Our premise is that each group is positively connected internally and negatively connected with the other $k-1$ groups. A distinguishing aspect of our formulation is that we are not searching for a complete partition of the signed network; instead, we allow a subset of nodes to be neutral with respect to the conflict structure we are searching. As a result, the problem we tackle differs from previously-studied problems, such as correlation clustering and $k$-way partitioning. To solve the conflicting-group discovery problem, we derive a novel formulation in which each conflicting group is naturally characterized by the solution to the maximum discrete Rayleigh's quotient (MAX-DRQ) problem. We present two spectral methods for finding approximate solutions to the MAX-DRQ problem, which we analyze theoretically. Our experimental evaluation shows that, compared to state-of-the-art baselines, our methods find solutions of higher quality, are faster, and recover ground-truth conflicting groups with higher accuracy.

## 1 Introduction

Signed networks are graphs where each edge is labeled either positive or negative. The introduction of edge signs, which goes back to the 50's, was motivated by the study of friendly and antagonistic social relationships [22]. The representation power of signed networks comes at the cost of significant differences in fundamental graph properties, and thus, algorithmic techniques employed to analyze unsigned networks are usually not directly applicable to their signed counterparts. These differences have spurred significant interest in a variety of analysis tasks in signed networks [20, 36] such as signed network embeddings [7, 24, 25, 39], signed clustering [8, 14, 27, 32], and signed link prediction [9, 28, 38, 41] in recent years.

In this paper we study the problem of detecting $k$ conflicting groups in signed networks. In more detail, we are interested in finding a collection of $k$ vertex subsets, each of which is positively connected internally, and negatively connected to the other $k-1$ subsets. In social networks where edge signs indicate positive or negative interactions, identifying conflicting groups may help in the study of polarization [1, 31, 40, 43], echo chambers [17, 19] and the spread of fake news [12, 35, 42].

Detecting $k$ conflicting groups is challenging due to various reasons. First, conflicting groups are not simply dense subgraphs, so community-detection techniques for unsigned graphs are not effective. Second, in real applications we can expect a majority of the network nodes to be neutral with respect

to the conflicting structure. As an example, consider a social network where a heated discussion is taking place between different political factions. Most users might not get involved in the quarrel, and thus their interactions are not necessarily consistent with this division. For this reason, methods for signed networks like correlation clustering and $k$-way partitioning may not be effective.

Our approach for detecting $k$-conflicting groups in signed networks extends the formulation of Bonchi et al. [4], to arbitrary values of $k$, addressing an open problem left in that work, which studied only the case $k = 2$. We argue that simply rounding the principal eigenvectors of the adjacency matrix might yield unsatisfactory results. Instead, we show that the proposed objective can be interpreted in terms of the Laplacian of a complete graph, and rely on the spectral properties of this matrix to derive a novel optimization framework, *spectral conflicting group detection* (SCG). By carefully examining the invariant subspaces of the aforementioned Laplacian, we reformulate the problem as a maximum discrete Rayleigh quotient (MAX-DRQ) objective, which is an **APX**-hard problem. We propose two algorithms, one deterministic, and one randomized with approximation guarantees. We show that the obtained approximation is essentially the best possible, when using the largest eigenvalue as an upper bound.

We perform an extensive set of experiments to compare the performance of our approach to that of multiple alternatives from the literature, on a variety of synthetic and real datasets. Our algorithms generally run faster, yield solutions of higher quality, and exhibit a better ability to find ground-truth groups than competing methods. In addition, we discuss how to select the number of groups $k$ in practical scenarios.

## 2  Related work

**Signed graph partition.** Typical formulations partition the nodes of the signed graph into $k$ sets so that intra-edges are mostly positive and inter-edges are mostly negative. This is a special case of $k$ conflicting group detection with no neutral nodes. Spectral methods are competitive and we review several representatives here. The signed Laplacian has been used for clustering [27], but resulting clusters tend to behave like in unsigned spectral clustering [37]. $k$-way balanced normalized cut (BNC) was proposed to address the issue [8]. Signed Laplacians [8, 27] were recently generalized through matrix power means [32]. The state of the art method SPONGE [14] is based on a generalized eigenvalue problem for constrained clustering [13] and works well on sparse graphs and large $k$. All these methods partition the network and are ineffective in the presence of many neutral nodes.

**Correlation clustering** methods partition the entire network, but allow $k$ to be unspecified. The standard objective [2, 5, 16, 6] counts the number of edges that agree (disagree) with the partition, i.e., positive (negative) intra-group edges and negative (positive) inter-group edges, and aims to maximize (minimize) agreement (disagreement). The problem is **APX**-hard for general graphs and has many variants. Giotis et al. [21] consider the case of fixed $k$ and Puleo et al. [33] measure per-node error. Our work is inspired by a recent variant [4], which formulates the discrete eigenvector problem by maximizing the gap between agreement and disagreement with respect to the total size of two conflicting groups. They propose a randomized $\mathcal{O}(\sqrt{n})$-approximation algorithm. However, their approach does not extend to $k > 2$, as the two groups are identified by the sign of the optimal vector. In fact, the discrete eigenvector problem is **APX**-hard and the best known result achieves an approximation guarantee of $\tilde{\mathcal{O}}(n^{1/3})$ using an SDP-based approach [3]. The latter SDP formulation cannot be extended to $k > 2$ as well. In this paper, we generalize the problem as MAX-DRQ and present two algorithms for $k \geq 2$.

**Antagonistic group mining** focuses on the setting with two groups. These works can be divided into direct or indirect. Direct methods [18, 29, 30] search for structures such as bi-cliques or balanced triads. Indirect methods [45, 46] find frequent conflicting patterns in database transactions. These approaches cannot be easily extended to finding $k > 2$ conflicting groups.

To our knowledge, our only direct competitor is the KOCG method [11]. They formulate the problem as trace-maximization, where each group is represented as a simplex with nonzero entries indicating the participation of the nodes in the groups. However, their method finds conflicting groups only within local regions and is sensitive to initialization, often converging to local maxima. Our approach is fundamentally different and experimentally is shown to consistently outperform this baseline.

# 3 Preliminaries

We focus on simple undirected signed graphs. We denote $G = (V, E)$ to be a signed graph, with $E = E_+ \cup E_-$ consisting of the sets of positive edges $E_+$ and negative edges $E_-$. The signed adjacency matrix of $G$ is denoted by $A \in \{-1, 0, 1\}^{n \times n}$ with $A_{i,j}$ being $+1$ if $(i, j) \in E_+$, $-1$ if $(i, j) \in E_-$ and 0 otherwise. We use $n = |V|$ and $m = |E|$ to indicate the number of nodes and edges of the signed graph $G$. We use $E(V_1, V_2)$ to denote the set of edges between two subsets $V_1, V_2 \subseteq V$, where $V_1, V_2$ are not required to be disjoint. We define $E(V_1)$ to be $E(V_1, V_1)$, for any $V_1 \subseteq V$.

We consider the eigenvalues $\lambda_1(M) \geq \ldots \geq \lambda_n(M)$ of a symmetric matrix $M \in \mathbb{R}^{n \times n}$, arranged in non-increasing order and listed with multiplicities. We denote the corresponding eigenvectors $\mathbf{v}_1(M), \ldots, \mathbf{v}_n(M)$, with $\mathbf{v}_i(M)$ associated with eigenvalue $\lambda_i(M)$. By convention, $\mathbf{v}_1(M)$ is the leading eigenvector and $\{\mathbf{v}_1(M), \ldots, \mathbf{v}_i(M)\}$ are the $i$ principal eigenvectors.

We denote by $I_n$ the identity matrix of size $n \times n$, and by $J_n$ the $n \times n$ matrix with all elements being 1. For a matrix $M \in \mathbb{R}^{n \times n}$, we use $M_{i,:}$ to indicate its $i$-th row, and $M_{:,j}$ to refer to its $j$-th column. We also use $M_{i:,j:}$ to indicate the submatrix of $M$ that consists of rows $i$ to $n$, and columns $j$ to $n$. We use $\text{tr}(\cdot)$ to denote the trace of a matrix, $\langle \cdot, \cdot \rangle_F$ to denote the Frobenius product between two matrices, and $\langle \cdot, \cdot \rangle$ to denote the dot product between two vectors. We use $\theta(\mathbf{u}, \mathbf{v}) = \arccos(\langle \mathbf{u}, \mathbf{v} \rangle / (\|\mathbf{u}\|_2 \|\mathbf{v}\|_2)) \in [0, \pi]$ to indicate the angle between two nonzero vectors $\mathbf{u}, \mathbf{v} \in \mathbb{R}^n$. Finally, we write $[n]$ to denote the set $\{1, \ldots, n\}$.

**Note.** All omitted proofs can be found in the supplementary material.

# 4 Problem formulation

Given a signed graph $G = (V, E)$ and an integer $k$, our goal is to find $k$ mutually-disjoint node sets $S_1, \ldots, S_k \subseteq V$ that have the following informally-stated properties:

**Property 1** *For all $i, j \in [k]$, with $i \neq j$, the edges in $E(S_i)$ are mostly positive, whereas the edges in $E(S_i, S_j)$ are mostly negative.*

**Property 2** *There should be a large number of interactions among the nodes of $S_1, \ldots, S_k$ relative to the total number of nodes in these groups. In other words, the subgraph induced by $S_1, \ldots, S_k$ should be as dense as possible.*

Inspired by the formulation of Bonchi et al. [4], our objective function is also a variant of the correlation-clustering problem [2], but with certain differences that we discuss below. For a set of groups $S_1, \ldots, S_k$ as a candidate solution, we quantify Property 1 by using the objective

$$f(S_1, \ldots, S_k) = \sum_{h \in [k]} \sum_{(i,j) \in E(S_h)} A_{i,j} + \frac{1}{k-1} \sum_{\substack{h, \ell \in [k] \\ h \neq \ell}} \sum_{(i,j) \in E(S_h, S_\ell)} (-A_{i,j}). \tag{1}$$

Compared to the standard objective of correlation clustering [2], which treats all edges equally, our objective in Equation (1) weighs an intra-group edge $k - 1$ times more heavily than an inter-group edge. The rationale is as follows: suppose the group sizes and edge densities stay fixed as $k$ increases. Since the number of inter-group edges grows quadratically with $k$ and the number of intra-group edges grows linearly with $k$, the weighting in Equation (1) prevents the inter-group edges from dominating the objective. The value of $(k - 1)$ in the denominator is chosen so that our objective reduces to the standard case, i.e., the formulation of Bonchi et al. [4], when $k = 2$.

By introducing an indicator matrix $X \in \{0, 1\}^{n \times k}$ with $X_{i,j} = 1$ if node $i \in S_j$ and 0 otherwise, our objective in Equation (1) can be rewritten as

$$f(S_1, \ldots, S_k) = \langle A, XX^T \rangle_F - \frac{\langle A, XJ_kX^T \rangle_F - \langle A, XX^T \rangle_F}{k-1} = \frac{\langle A, XL_kX^T \rangle_F}{k-1}, \tag{2}$$

where $L_k = kI_k - J_k$. The term $XL_kX^T$ in Equation (2) captures explicitly the relationship between the $k$ groups as $(XL_kX^T)_{i,j}$ is positive (negative) whenever nodes $i$ and $j$ are in the same (different) groups. Also, $(XL_kX^T)_{i,j} = 0$ if either node $i$ or node $j$ does not belong to any of the groups.

Hence, the value of the Frobenius product $\langle A, XL_kX^T\rangle_F$ quantifies Property (1) for the groups $S_1, \ldots, S_k$ (which are encoded in matrix $X$).

Next, we analyze the matrix $L_k$, which is a fixed matrix not depending on the input signed graph. Let $L_k = UDU^T$ be the eigendecomposition of $L_k$, where $D = \mathrm{diag}([0, k, \ldots, k]) \in \mathbb{R}^{k \times k}$, and $U \in \mathbb{R}^{k \times k}$ is a real-valued orthogonal matrix. As the geometric multiplicity of eigenvalue $k$ is $k - 1$, the matrix $U$ is not unique. For the rest of the paper, we restrict our choice of $U$ to be the following

$$
\begin{aligned}
(U_{:,1})^T &= 1/\sqrt{k}\,[1, \ldots, 1], & (U_{:,2})^T &= c_1\,[k - 1, -1, \ldots, -1], \\
(U_{:,3})^T &= c_2\,[0, k - 2, -1, \ldots, -1], & \ldots \quad (U_{:,k})^T &= c_{k-1}\,[0, \ldots, 0, 1, -1],
\end{aligned}
\tag{3}
$$

where $c_i = 1/\sqrt{(k - i + 1)(k - i)}$, for $i = 1, \ldots, k - 1$.

By the change of variables $Y = XU$, we can rewrite our objective in Equation (2) as

$$
\langle A, XL_kX^T\rangle_F = \langle A, Y\,\mathrm{diag}([0, k, \ldots, k])Y^T\rangle_F = k\,\mathrm{tr}((Y_{:,2:})^T A (Y_{:,2:})).
\tag{4}
$$

To account for Property (2) we normalize our objective with the total number of nodes in the groups $S_1, \ldots, S_k$, which can be written as

$$
\sum_{i \in [k]} |S_i| = \mathrm{tr}(Y^T Y) = k\,(Y_{:,1})^T(Y_{:,1}) = \frac{k}{k - 1}\,\mathrm{tr}((Y_{:,2:})^T(Y_{:,2:})).
\tag{5}
$$

Finally we replace the constraints on the indicator matrix $X$ with the constraint that the rows of $Y$ should take values in the set $\{\mathbf{0}, U_{1,:}, \ldots, U_{k,:}\}$. The equivalence holds since $X_{i,:}$ picks the $j$-th row of $U$ if $i \in S_j$. Putting all this together, we can now give the final formulation of our problem:

$$
\max_{Y \in \mathbb{R}^{n \times k} \setminus \{\mathbf{0}\}} \frac{\mathrm{tr}((Y_{:,2:})^T A (Y_{:,2:}))}{\mathrm{tr}((Y_{:,2:})^T (Y_{:,2:}))},
\tag{6}
$$

$$
\text{subject to} \quad Y_{i,:} \in \{\mathbf{0}, U_{1,:}, \ldots, U_{k,:}\}, \text{ for all } i = 1, \ldots, n.
$$

Intuitively, our objective aims to find small-size conflicting groups with many edges satisfying Property (1). Note that if we ignore the weighting between the inter-group and intra-group edges, Equation (6) can be expressed as $(\#\{\text{edges satisfying Property (1)}\} - \#\{\text{edges violating Property (1)}\})$ divided by $|\cup_{h \in [k]} S_h|$.

Also, note that our optimization problem, as formulated above, is different from the trace-maximization problem [26], which given two $n \times n$ matrices $M$ and $A$, seeks to find an $n \times d$ matrix $Z$ to maximize the form $\mathrm{tr}(Z^T A Z)$, subject to the constraint $Z^T M Z = I_d$. The reason is that since we have no constraint on the group sizes, there is no predefined matrix $M$ to require $X^T M X = I_k$.

# 5 Proposed spectral approach

The problem we study has been shown to be **APX**-hard for the special case of $k = 2$ [3]. Here we consider a generalization for any $k \geq 2$. In this section we present an efficient spectral algorithm by leveraging the problem formulation (6).

Our starting point is that matrix $U$, as seen in Equations (3), is almost lower-triangular. We can use this observation to partition $Y_{:,2:}$ column-wise, and reformulate the constraints in problem formulation (6) as follows:

$$
Y_{:,2} \in \{0, -c_1, c_1(k - 1)\}^n \text{ implies } Y_{i,2} = \begin{cases} c_1(k - 1) & \text{if } i \in S_1, \\ -c_1 & \text{if } i \in \cup_{h=2}^{k} S_h, \end{cases}
$$

$$
Y_{:,3} \in \{0, -c_2, c_2(k - 2)\}^n \text{ implies } Y_{i,3} = \begin{cases} 0 & \text{if } i \in S_1, \\ c_2(k - 2) & \text{if } i \in S_2, \\ -c_2 & \text{if } i \in \cup_{h=3}^{k} S_h, \end{cases}
$$

$$
\vdots
$$

$$
Y_{:,k} \in \{0, -c_{k-1}, c_{k-1}\}^n \text{ implies } Y_{i,k} = \begin{cases} 0 & \text{if } i \in \cup_{h=1}^{k-2} S_h, \\ c_{k-1} & \text{if } i \in S_{k-1}, \\ -c_{k-1} & \text{if } i \in S_k. \end{cases}
$$

---

**Algorithm 1:** SCG $(A, k)$                                           `Spectral Conflicting Group detection`

---

**Input** : $A$ is the adjacency matrix of the signed network; $k$ is the number of groups.
**Output** : Groups $S_1, \ldots, S_k$.
$A^{(0)} \leftarrow A$;
**for** $t = 1, \ldots, k - 1$ **do**
> $\mathbf{r}^{(t)} \leftarrow$ Solve-Max-DRQ $(A^{(t-1)}, k - t)$ ;                          `// See Algorithm 2`
> **if** $t < k - 1$ **then**
>> $S_t \leftarrow \{i \notin \cup_{j=1}^{t-1} S_j : |\mathbf{r}_i^{(t)}| = (k - t)\}$;
>> $A^{(t)} \leftarrow A^{(t-1)}$;
>> $A_{i,:}^{(t)} \leftarrow \mathbf{0}_{1 \times n}$ and $A_{:,i}^{(t)} \leftarrow \mathbf{0}_{n \times 1}$ for all $i \in S_t$ ;              `// Remove edges` $E(S_t, V)$
> **else** $S_{k-1} \leftarrow \{i \notin \cup_{j=1}^{t-1} S_j : \mathbf{r}_i^{(t)} = 1\}$ and $S_k \leftarrow \{i \notin \cup_{j=1}^{t-1} S_j : \mathbf{r}_i^{(t)} = -1\}$ ;
**end**
**return** $S_1, \ldots, S_k$;

---

Notice that $Y_{i,j} = 0$ for all $i \in \cup_{h=1}^{j-2} S_h$ and $Y_{i,j} = -c_{j-1}$ for all $i \in \cup_{h=j}^{k} S_h$. We let $A^{(0)} = A$, and we define $A^{(t)}$ to be the adjacency matrix that results after removing from $A^{(t-1)}$ all entries that correspond to edges incident to nodes in $S_t$. Then, the objective function (6) is equivalent to

$$\frac{\text{tr}((Y_{:,2:})^T A (Y_{:,2:}))}{\text{tr}((Y_{:,2:})^T (Y_{:,2:}))} = \sum_{t=1}^{k-1} w_t \frac{(Y_{:,t+1})^T A (Y_{:,t+1})}{(Y_{:,t+1})^T (Y_{:,t+1})} = \sum_{t=1}^{k-1} w_t \frac{(Y_{:,t+1})^T A^{(t-1)} (Y_{:,t+1})}{(Y_{:,t+1})^T (Y_{:,t+1})}, \quad (7)$$

where $w_t = (Y_{:,t+1})^T (Y_{:,t+1}) / \text{tr}((Y_{:,2:})^T (Y_{:,2:})) \in [0, 1]$ and $\sum_{t=1}^{k-1} w_t = 1$. In other words, Equation (7) shows that the objective function (6) is a convex combination of $k - 1$ discrete Rayleigh quotients. Moreover, Equation (7) also suggests that the solution $Y_{:,t+1}$ characterizes the group $S_t$ that conflicts the most with the (not yet decided) rest of groups $S_h$ for $h > t$. Based on this observation, we propose a scheme SCG (spectral conflicting groups), shown as Algorithm 1.

SCG executes $k - 1$ iterations. At the $t$-th iteration, for each $t \in [k-1]$, we find the vector $Y_{:,t+1}$ that maximizes the discrete Rayleigh quotient of $A^{(t-1)}$, while satisfying the constraints set on matrix $Y$. We refer to this problem as MAX-DRQ:

$$\mathbf{r}^{(t)} = \underset{\mathbf{x} \in \{0, -1, k-t\}^n \setminus \{\mathbf{0}\}}{\text{argmax}} \frac{\mathbf{x}^T A^{(t-1)} \mathbf{x}}{\mathbf{x}^T \mathbf{x}}. \quad (8)$$

The vector $Y_{:,t+1}$ is then given by $Y_{:,t+1} = c_t \mathbf{r}^{(t)}$. We note that our scheme works with any method that solves the MAX-DRQ problem. In Algorithm 1 (SCG) we refer to such a general method as Solve-Max-DRQ. Strategies to solve MAX-DRQ are presented in Section 6. Once the MAX-DRQ problem is solved in the $t$-th iteration, the vector $\mathbf{r}^{(t)}$ is obtained. If $t < k - 1$, the $t$-th group is recovered by $S_t = \{i \notin \cup_{j=1}^{t-1} S_j : |\mathbf{r}_i^{(t)}| = (k-t)\}$, and if $t = k-1$ (last iteration), the last two groups are recovered by $S_{k-1} = \{i \notin \cup_{j=1}^{t-1} S_j : \mathbf{r}_i^{(t)} = 1\}$ and $S_k = \{i \notin \cup_{j=1}^{t-1} S_j : \mathbf{r}_i^{(t)} = -1\}$.

Note that Equation (7) justifies why it is not a good idea to use the $k - 1$ principal vectors of $A$ to identify the conflicting groups: the reason is that the coefficients $[w_t]$ are not fixed values.

## 6  Solving the maximum discrete Rayleigh quotient problem

In this section we present two solutions for MAX-DRQ. Our first solution is a deterministic algorithm presented in Section 6.1. The second solution is a randomized algorithm presented in Section 6.2. Both solutions first compute the leading eigenvector $\mathbf{v}_1$ of the input matrix $A^{(t-1)}$, and then round $\mathbf{v}_1$ to the appropriate discrete form. The difference is the rounding method. We refer to this generic algorithm as Solve-Max-DRQ, and it is the procedure used in the iterative step of SCG. Pseudocode for Solve-Max-DRQ is given as Algorithm 2.

---

**Algorithm 2:** Solve-Max-DRQ $(A, q)$            `Find maximum discrete Rayleigh quotient`

---

**Input** : Square and symmetric matrix $A$, and positive integer $q$.
**Output :** The rounded vector $\mathbf{r} \in \{0, -1, q\}^n$.
$\mathbf{v} \leftarrow$ the leading eigenvector of $A$;
$(d_1, \mathbf{r}_1) \leftarrow$ Round$(\mathbf{v}, q)$ ;                             `//` $d_1 = \sin\theta(\mathbf{v}, \mathbf{r}_1)$
$(d_2, \mathbf{r}_2) \leftarrow$ Round$(-\mathbf{v}, q)$ ;                         `//` $d_2 = \sin\theta(-\mathbf{v}, \mathbf{r}_2)$
**if** $d_1 \leq d_2$ **then** $\mathbf{r} \leftarrow \mathbf{r}_1$;
**else** $\mathbf{r} \leftarrow \mathbf{r}_2$;
return $\mathbf{r}$;

---

---

**Algorithm 3:** MinAngleRound $(\mathbf{v}, q)$     `Deterministic rounding by minimum-angle heuristic`

---

**Input** : Vector $\mathbf{v} \in \mathbb{R}^n$ and positive integer $q$.
**Output :** Vector $\mathbf{u}^* \in \{0, -1, q\}^n$ with min angle to $\mathbf{v}$.
$\{i_k\}_{k=1}^n \leftarrow$ Sort $\mathbf{v}$ and return the indexes such that $\mathbf{v}_{i_1} \geq \ldots \geq \mathbf{v}_{i_n}$;
$(d, \mathbf{u}^*) \leftarrow (\infty, \mathbf{0})$;
$(k_1, k_2) \leftarrow (0, n+1)$;
**while** $k_1 < k_2$ **do**
    $\mathbf{u}_1 \leftarrow$ set the $i_{k_1+1}$-th element of $\mathbf{u}^*$ to $q$;
    $\mathbf{u}_2 \leftarrow$ set the $i_{k_2-1}$-th element of $\mathbf{u}^*$ to $-1$;
    **if** $\min\{\sin\theta(\mathbf{v}, \mathbf{u}_1), \sin\theta(\mathbf{v}, \mathbf{u}_2)\} \geq d$ **then** break;
    **if** $\sin\theta(\mathbf{v}, \mathbf{u}_1) < \sin\theta(\mathbf{v}, \mathbf{u}_2)$ **then** $(k_1, d, \mathbf{u}^*) \leftarrow (k_1 + 1, \sin\theta(\mathbf{v}, \mathbf{u}_1), \mathbf{u}_1)$;
    **else** $(k_2, d, \mathbf{u}^*) \leftarrow (k_2 - 1, \sin\theta(\mathbf{v}, \mathbf{u}_2), \mathbf{u}_2)$;
**end**
return $(d, \mathbf{u}^*)$;

---

## 6.1 Deterministic rounding

Our goal is to find a discrete vector $\mathbf{v}^* \in \{0, -1, q\}^n$ that maximizes the quotient $\mathbf{x}^T A^{(t-1)} \mathbf{x}/(\mathbf{x}^T \mathbf{x})$. Let $\mathbf{v}$ be the leading eigenvector of $A^{(t-1)}$, i.e., the real-valued maximizer of $\mathbf{x}^T A^{(t-1)} \mathbf{x}/(\mathbf{x}^T \mathbf{x})$. The idea is to round $\mathbf{v}$ to a discrete vector $\mathbf{u}^* \in \{0, -1, q\}^n$ that minimizes $\sin\theta(\mathbf{v}, \mathbf{u})$, among all vectors $\mathbf{u} \in \{0, -1, q\}^n$. It can be shown that such $\mathbf{u}^*$ can be found by restricting the search over $\mathcal{O}(n^2)$ thresholded candidate vectors obtained by $\mathbf{v}$. We formalize this below.

**Definition 1** *Let $\mathbf{v} \in \mathbb{R}^n$, $q \in [k-1]$ and $a, b \in \mathbb{R}$ be given. Define a threshold function $\sigma_{a,b} : \mathbb{R}^n \to \mathbb{R}^n$ that maps $\mathbf{v}$ to a new vector $\sigma_{a,b}(\mathbf{v})$, whose $i$-th coordinate is*

$$
\sigma_{a,b}(\mathbf{v})_i = \begin{cases} q & \text{if } \mathbf{v}_i \geq a > 0, \\ -1 & \text{if } \mathbf{v}_i \leq b < 0, \\ 0 & \text{otherwise,} \end{cases}
$$

*and denote $\mathcal{T} = \{t_i\}_{i=0}^{n+1}$ the sequence of all possible thresholds over the coordinates of $\mathbf{v}$, that is, $t_0 = \infty$, $t_{n+1} = -\infty$ and $t_i$ is the $i$-th largest coordinate of $\mathbf{v}$, for $i \in [n]$. Then, the set of all possible thresholded vectors of $\mathbf{v}$ is denoted by $\Gamma(\mathbf{v}) = \{\sigma_{a,b}(\mathbf{v}) : \text{for all } a, b \in \mathcal{T}\}$.*

Given a vector $\mathbf{v}$, the discrete vector $\mathbf{u}^* \in \{0, -1, q\}^n$ that minimizes $\sin\theta(\mathbf{v}, \mathbf{u})$ can be computed by using the following result.

**Lemma 1** *Let $\mathbf{v} \in \mathbb{R}^n$ and $q \in [k-1]$ be given. The minimizer of $\sin\theta(\mathbf{v}, \mathbf{u})$ over all $\mathbf{u} \in \{0, -1, q\}^n$ is equal to the minimizer of $\sin\theta(\mathbf{v}, \mathbf{u})$ over all $\mathbf{u} \in \Gamma(\mathbf{v}) \cup \Gamma(-\mathbf{v})$.*

Since the size of the set $\Gamma(\mathbf{v}) \cup \Gamma(-\mathbf{v})$ is $\mathcal{O}(n^2)$, enumerating all vectors to find the optimal $\mathbf{u}$ is not efficient for large datasets. To make our method scalable, we propose a linear-time rounding heuristic in Algorithm 3, which finds a local optimum.

This heuristic works by initializing two indexes $k_1, k_2$, the indexes of the two thresholds, which are initially set to 0 and $n + 1$, respectively. At each iteration, we move only 1 threshold, we either increase $k_1$ by 1 or decrease $k_2$ by 1. This is determined by comparing $\sin\theta(\mathbf{v}, \sigma_{t_{k_1+1}, t_{k_2}}(\mathbf{v}))$ and $\sin\theta(\mathbf{v}, \sigma_{t_{k_1}, t_{k_2-1}}(\mathbf{v}))$ and choosing the smaller option.

---
**Algorithm 4:** RandomRound $(\mathbf{v}, q)$                                              `Randomized rounding`
---
**Input**   : Vector $\mathbf{v} \in \mathbb{R}^n$ and positive integer $q$.
**Output:** Vector $\mathbf{u} \in \{0, -1, q\}^n$, a randomized rounded vector of $\mathbf{v}$.
$\mathbf{u} \leftarrow \mathbf{0}$;
**for** $i = 1, \ldots, n$ **do**
    **if** $\mathbf{v}_i > 0$ **then**  $\mathbf{u}_i \leftarrow q \, \mathrm{Bernoulli}(|\mathbf{v}_i|/q)$ ;
    **else if** $\mathbf{v}_i < 0$ **then**  $\mathbf{u}_i \leftarrow (-1) \, \mathrm{Bernoulli}(|\mathbf{v}_i|)$ ;
**end**
$d \leftarrow \sin\theta(\mathbf{v}, \mathbf{u})$;
return $(d, \mathbf{u})$;
---

### 6.2 Randomized rounding

Our second algorithm for maximizing $\mathbf{x}^T A^{(t-1)}\mathbf{x}/(\mathbf{x}^T\mathbf{x})$ in $\{0, -1, q\}^n$ is a randomized-rounding scheme starting with the eigenvector $\mathbf{v}$ of $A^{(t-1)}$. Pseudocode is shown in Algorithm 4.

In more detail, we round $\mathbf{v}$ onto $\{0, -1, q\}^n$ by drawing Bernoulli trials. For each positive coordinate $\mathbf{v}_i$ we set $\mathbf{u}_i \sim q \, \mathrm{Bernoulli}(|\mathbf{v}_i|/q)$, for each negative coordinate $\mathbf{v}_i$ we set $\mathbf{u}_i \sim (-1) \, \mathrm{Bernoulli}(|\mathbf{v}_i|)$, and if $\mathbf{v}_i = 0$ we set $\mathbf{u}_i = 0$. In this way, we have $\mathbb{E}[\mathbf{u}] = \mathbf{v}$. By applying similar arguments to the ones presented by Bonchi et al. [4], we can show that the randomized-rounding algorithm provides a $\mathcal{O}(q\sqrt{n})$-approximation guarantee to the MAX-DRQ problem. We present this result as Theorem 1. Furthermore, Corollary 1 states that this result is tight for $k = 2$.

**Theorem 1** *Let $\mathbf{v}$ be the leading eigenvector of the adjacency matrix $A$ of a signed graph, and let $q \geq 1$ be a positive integer. Then, the* RandomRound *algorithm with $(\mathbf{v}, q)$ as input is a $(q\sqrt{n})$-approximation to the optimum of the corresponding* MAX-DRQ *problem.*

**Lemma 2** *Let $OPT$ be the optimum solution to the* MAX-DRQ *problem. There exists a problem instance such that $\lambda_1(A) \geq OPT \cdot \Omega(\sqrt{n})$.*

**Corollary 1** *The integrality gap of algorithm* RandomRound *is $\Omega(\sqrt{n})$, and thus, the approximation result of Theorem 1 is asymptotically tight up to a factor of $q$.*

## 7 Experimental evaluation

In this section, we evaluate our framework with both synthetic and real-world graphs. All the experiments are executed on a machine with Intel Core i5 at 1.8 GHz with 8 GB RAM. All methods have been implemented in Python 3.[1] The datasets we have used are all publicly available and the detailed information can be found in Supplementary § D.1. Beyond the experiments discussed here, we present more results in Supplementary § D, including execution times, and a discussion on deciding the number of groups $k$.

**Proposed methods.** Our approach (SCG) is a framework that admits different methods to solve MAX-DRQ. We have instantiated our framework with the following routines. *Minimum angle*: the deterministic rounding algorithm presented in Section 6.1; *Randomized rounding*: the randomized rounding algorithm presented in Section 6.2; *Maximum objective*: a generalization of EigenSign [4], that rounds $\mathbf{v}_1(A)$ by finding an optimal threshold to maximize the objective; *Bansal*: motivated by the pivot for correlation clustering [2], which finds two conflicting groups by considering the neighborhood of a single node, and using the node that results in the maximum value of the objective. These instantiations are denoted by SCG-MA, SCG-R, SCG-MO, and SCG-B, respectively.

**Baselines.** We use the following baselines: KOCG [11] is a method designed for a similar formulation to ours. We use the authors' implementation [10] with default hyperparameters $\alpha = 1/(k-1)$, $\beta = 50$, and $\ell = 5000$. As KOCG returns a ranked list of disjoint subgraphs, each containing $k$ conflicting groups, we pick the $k$ groups contained in their *top-1* and *top-r* subgraphs. We choose $r$ so that the total group size equals the one returned by SCG-MA. We use two spectral algorithms: BNC [8], which optimizes *balanced normalized cut*; and SPONGE [14], a method particularly suitable

Table 1: Polarity objective (Equation (6)) achieved by the proposed methods and the baselines on real-world signed graphs, for two different values of $k$: the number of conflicting groups to be detected. Dashes indicate that a method exceeded the memory limit.

| | | WoW-EP8 | Bitcoin | WikiVot | Referendum | Slashdot | WikiCon | Epinions | WikiPol |
|---|---|---|---|---|---|---|---|---|---|
| | $|V|$ | 790 | 5 881 | 7 115 | 10 884 | 82 140 | 116 717 | 131 580 | 138 587 |
| | $|E|$ | 116 009 | 21 492 | 100 693 | 251 406 | 500 481 | 2 026 646 | 711 210 | 715 883 |
| | $|E_-|/|E|$ | 0.2 | 0.2 | 0.2 | 0.1 | 0.2 | 0.6 | 0.2 | 0.1 |
| $k=2$ | SCG-MA | **236.6** | **28.8** | **71.5** | 172.2 | 77.5 | 155.2 | 128.3 | **82.8** |
| | SCG-MO | **236.6** | **29.5** | **71.7** | **174.1** | **79.7** | **175.7** | **128.7** | **88.4** |
| | SCG-B | 200.6 | 21.7 | 37.6 | 116.3 | 61.0 | 129.3 | **156.4** | 46.5 |
| | SCG-R | 218.3 | 14.9 | 55.7 | 119.6 | 29.9 | 100.2 | 70.9 | 36.0 |
| | KOCG-top-1 | 9.0 | 3.6 | 4.0 | 4.3 | 1.0 | 6.2 | 4.2 | 1.0 |
| | KOCG-top-$r$ | 18.2 | 3.8 | 2.5 | 14.0 | 3.7 | 2.4 | 6.2 | 0.9 |
| | BNC-$k$ | 184.6 | 5.3 | 15.8 | 41.5 | — | — | — | — |
| | BNC-$(k+1)$ | -0.7 | -10.8 | -1.0 | -1.0 | — | — | — | — |
| | SPONGE-$k$ | 191.4 | 5.1 | 15.8 | 41.5 | — | — | — | — |
| | SPONGE-$(k+1)$ | 88.0 | 1.0 | 1.0 | 1.0 | — | — | — | — |
| $k=6$ | SCG-MA | 207.3 | **14.6** | **45.5** | 84.9 | 37.8 | 102.6 | 88.8 | **57.5** |
| | SCG-MO | **226.9** | **15.2** | **47.0** | 55.6 | 34.6 | **111.6** | **129.2** | 41.8 |
| | SCG-B | **211.6** | 9.3 | 23.3 | **116.2** | **47.7** | 46.1 | **94.5** | **46.0** |
| | SCG-R | 198.1 | 5.0 | 9.7 | 39.8 | 7.3 | 16.2 | 39.4 | 5.5 |
| | KOCG-top-1 | 7.0 | 4.4 | 5.5 | 8.8 | 2.6 | 4.5 | 8.7 | 4.8 |
| | KOCG-top-$r$ | 8.5 | 2.9 | 2.9 | 5.0 | 3.6 | 4.0 | 6.5 | 3.0 |
| | BNC-$k$ | 185.2 | 5.2 | 15.8 | 41.5 | — | — | — | — |
| | BNC-$(k+1)$ | -0.2 | -4.2 | -1.1 | -0.8 | — | — | — | — |
| | SPONGE-$k$ | 58.5 | 5.0 | 15.8 | 41.5 | — | — | — | — |
| | SPONGE-$(k+1)$ | 48.1 | 0.8 | 1.0 | 1.0 | — | — | — | — |

for sparse graphs and large $k$. To detect $k$ conflicting groups using the spectral clustering algorithms, we compare with two approaches. The first approach is to directly apply BNC and SPONGE to detect $k$ clusters and return all the detected clusters as conflicting groups. The second approach is to detect $(k+1)$ clusters, then heuristically treat the largest cluster as the non-conflicting cluster, and return the $k$ smallest clusters as the detected conflicting groups. Let BNC-$k$ and SPONGE-$k$ denote SPONGE and BNC with the first approach and let BNC-$(k+1)$ and SPONGE-$(k+1)$ denote the two with the second approach. We use a publicly-available implementation [15] for BNC and SPONGE.

**Results on real-world networks.** We first measure the quality of the proposed methods and baselines with respect to the polarity objective, i.e., Equation (6), on real-world signed graphs. The results are shown in Table 1. The running times of all methods are listed in Supplementary § D.2. We observe that mostly, SCG-MA and SCG-MO achieve the best polarity scores. They are also the fastest, and usually find larger groups. An example of the sizes of the groups found by all methods is given in Supplementary § D.3. The SCG-B algorithm identifies conflicting groups by exploring local neighborhoods, and its detected groups tend to be located around high-degree nodes. Although SCG-B achieves the largest polarity on `Referendum` for $k=6$, it only detects 2 groups, already covered by SCG-MA and SCG-MO. As the groups are not necessarily the high-degree nodes, SCG-B performs less competitive on `WikiVot` and `WikiCon` for $k=6$. Finally, SCG-R is not as competitive as SCG-MA or SCG-MO and is slower due to random sampling.

With respect to our direct competitor KOCG, the KOCG-top-1 variant performs slightly better than KOCG-top-$r$ when $k=6$. As KOCG finds groups in local regions, KOCG-top-1 returns much smaller groups than the other methods. On the contrary, KOCG-top-$r$ intersects several local groups in different graph regions but remains ineffective compared to SCG-MA under the same total group size. All KOCG settings perform worse than BNC and SPONGE on the first 4 datasets.

Finally, the spectral-clustering methods BNC and SPONGE exceed the memory limit (caused by $k$-means) on large datasets. The $k$ groups returned by BNC-$k$ and SPONGE-$k$ usually consist of one large group with many non-conflicting nodes and $k-1$ very small groups. Since BNC-$(k+1)$ and SPONGE-$(k+1)$ can use the spare cluster to put the non-conflicting nodes, we expect they perform better than BNC-$k$ and SPONGE-$k$ but it turns out to be worse on all 4 real-world networks. Despite of the unexpected results, both versions of BNC and SPONGE are less effective than SCG-MA and SCG-MO at finding conflicting groups in real-world graphs.

**Results on synthetic graphs.** In our second experiment, we use synthetic graphs to measure how well the methods recover ground-truth conflicting groups. We use the *modified signed stochastic*

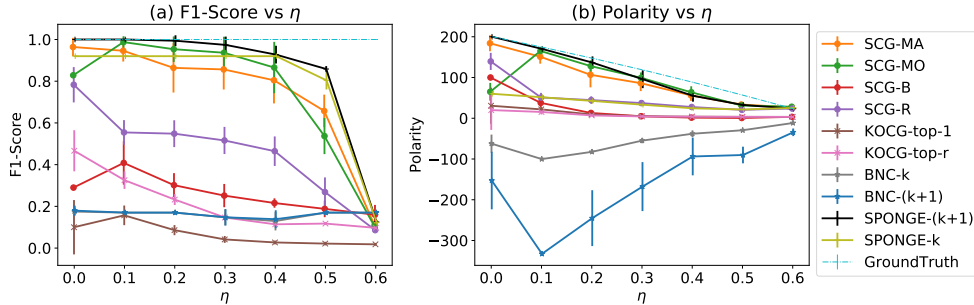

Figure 1: $F_1$-score (left) and polarity score (right) as a function of the parameter $\eta$. The input signed graphs are generated by the m-SSBM model, for a graph of size $n = 2\,000$, with $k = 6$ ground-truth groups, each having size $\ell = 100$.

*block model* (m-SSBM) [4], which has 4 parameters; $n$: the graph size; $k$: the number of conflicting groups; $\ell$: the size of each of the conflicting groups (all have the same size); and $\eta \in [0, 1]$: a parameter that controls the edge probabilities. Edges in the same group are positive with probability $1 - \eta$ and negative or absent with probability $\eta/2$. Edges between distinct groups are negative with probability $1 - \eta$ and positive or absent with probability $\eta/2$. All other edges have equal probability of $\min(\eta, 1/2)$ of being positive or negative. Hence, the smaller the value of $\eta$, the denser the conflicting groups and the lower the noise level. Note that the conflicting groups only emerge when $\eta \leq 2/3$, since m-SSBM is expected to have more negative edges in the groups and more positive edge between groups if $\eta > 2/3$.

In this experiment we measure the recovery rate of the ground-truth groups using the $F_1$ score, with precision and recall averaged over all groups. In Figure 1 we report the results of the m-SSBM model with parameters $n = 2\,000$, $k = 6$, $\ell = 100$, and $\eta = 0 : 0.1 : 0.6$. Each setting is repeated 20 times, and we report the average $F_1$ score and polarity scores.

As seen in Figure 1, the recovery rate ($F_1$ score) for all methods declines with $\eta$, since the graph becomes sparser and more noisy. It is clear that SCG-MA and SCG-MO are robust methods, handling very well the increasing noise level. It is worth noting that SPONGE-$(k + 1)$ performs the best in this experiment with respect to both $F_1$ and polarity. We also see that SCG-B is less competitive here, as in this data the conflicting groups are not concentrated around high-degree nodes. In summary, under the m-SSBM model, our polarity score is consistent with the $F_1$ score, and our proposed methods SCG-MA and SCG-MO are effective in detecting the ground-truth conflicting groups.

## 8   Conclusions and future work

We propose an efficient method for detecting $k$ conflicting groups in a signed network. Our approach relies on interpreting the problem objective in terms of the Laplacian of a complete graph, characterizing the spectral properties of this matrix, and deriving a novel formulation in which each conflicting group is characterized by the solution to the maximum discrete Rayleigh quotient problem.

Our work opens several exciting directions for future work. First, it remains open whether we can improve the $\mathcal{O}(\sqrt{n})$-approximation for the maximum discrete Rayleigh quotient problem, using an approach that does not rely on rounding the leading eigenvector, such as by extending the SDP-based algorithm in [3]. Second, it would be interesting to explore the applicability of our approach to unsigned graphs for the task of detecting dense subgraphs. Third, the modified Stochastic Block Model (m-SSBM) is actually a special case of Label Stochastic Block Model (LSBM) [23]. It would be relevant to analyze the recovery guarantee of our proposed method in m-SSBM with respect to the fundamental limit results [44] and the interplay with the Bethe-Hessian operator [34] in the sparse regime. Finally, the difference in the empirical performance of our two rounding techniques and the spectral clustering baseline SPONGE [14] in the real-world networks and the synthetic network is somewhat striking. It is possible that some properties or structures exist in the real-world networks but not in the synthetic networks. An interesting question is to explain this behavior analytically, in particular with respect to properties of real-world networks.

## Acknowledgments

We thank the anonymous reviewers for their insightful feedback. We thank Stefan Neumann for pointing to us the work of Bhaskara et al. [3]. This research is supported by the Academy of Finland projects AIDA (317085) and MLDB (325117), the ERC Advanced Grant REBOUND (834862), the EC H2020 RIA project SoBigData (871042), and the Wallenberg AI, Autonomous Systems and Software Program (WASP) funded by the Knut and Alice Wallenberg Foundation.

## Broader Impact

As the task we tackle in this paper belongs to the broad category of data mining, and as our study is mainly of theoretical nature, the impact of our work to the society is indirect. With respect to positive consequences, we name two possible applications that could impact the modern society. First, the rise of polarization and fake news is related to the existence of conflicting groups. Thus, having an efficient characterization tool is the first step to mitigate the situation. Second, both collaboration and competition exist in a diverse environment and detecting conflicting groups helps to understand the interplay of the two. With respect to negative consequences, we do not foresee specific issues when applying our method.

## Footnotes

[1]https://github.com/rutzeng/SCG-NeurIPS2020.

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
