[Supplementary Material]

# Discovering conflicting groups in signed networks
# Supplementary material

**Ruo-Chun Tzeng**
KTH Royal Institute of Technology
rctzeng@kth.se

**Bruno Ordozgoiti**
Aalto University
bruno.ordozgoiti@aalto.fi

**Aristides Gionis**
KTH Royal Institute of Technology
argioni@kth.se

## A    Proof of Lemma 1

**Proof:**    We have $\|\mathbf{v}\|_2 = 1$ and, without loss of generality, we can assume that the coordinates of $\mathbf{v}$ are sorted in non-increasing order. Let $\mathcal{T} = \{t_i\}_{i=0}^{n+1}$ be all possible thresholds for $\mathbf{v}$ and $\mathcal{T}' = \{t_i'\}_{i=0}^{n+1}$ be all possible thresholds for $-\mathbf{v}$. Recall the definition of $\theta(\cdot, \cdot)$ from Section 3 that $\theta(\mathbf{a}, \mathbf{b}) = \arccos(\langle \mathbf{a}, \mathbf{b} \rangle / \|\mathbf{a}\|_2 \|\mathbf{b}\|_2) \in [0, \pi]$ for any two nonnegative vectors $\mathbf{a}, \mathbf{b} \in \mathbb{R}^n$, so $\sin \theta(\mathbf{a}, \mathbf{b})$ is always non-negative. Let $\mathbf{u}^*$ be the minimizer of $\sin \theta(\mathbf{v}, \mathbf{u})$ over all $\mathbf{u} \in \Gamma(\mathbf{v}) \cup \Gamma(-\mathbf{v})$.

For simplicity, we assume $\mathbf{u}^* \in \Gamma(\mathbf{v})$ and $\langle \mathbf{v}, \mathbf{u}^* \rangle \geq 0$. This is because if the dot product is negative, we can make it positive by reversing the sign of $\mathbf{v}$. Let $k_1^*, k_2^*$ be the two thresholds such that $\mathbf{u}^* = \sigma_{t_{k_1^*}, t_{k_2^*}}(\mathbf{v})$. We will show that $\sin \theta(\mathbf{v}, \mathbf{u}) \geq \sin \theta(\mathbf{v}, \mathbf{u}^*)$ for any $\mathbf{u} \in \{0, -1, q\}^n$.

Fix any $\mathbf{u} \in \{0, -1, q\}^n$. Our first step is to identify the coordinates that $\mathbf{u}_i \neq \mathbf{u}_i^*$, denoted by $\mathcal{I} = \{j : \mathbf{u}_j \neq \mathbf{u}_j^*\}$. Moreover, since $\mathbf{u}_j^* = q$ for all $j \leq k_1^*$, $\mathbf{u}_j^* = -1$ for all $j \geq k_2^*$, and $\mathbf{u}_j^* = 0$ for all $j \in (k_1^*, k_2^*)$, we further divide $\mathcal{I}$ into 6 disjoint subsets:

$$\mathcal{I}_{11} = \{j \in \mathcal{I} : \mathbf{u}_j = 0, j \leq k_1^*\}, \qquad \mathcal{I}_{12} = \{j \in \mathcal{I} : \mathbf{u}_j = -1, j \leq k_1^*\},$$
$$\mathcal{I}_{21} = \{j \in \mathcal{I} : \mathbf{u}_j = q, j \in (k_1^*, k_2^*)\}, \qquad \mathcal{I}_{22} = \{j \in \mathcal{I} : \mathbf{u}_j = -1, j \in (k_1^*, k_2^*)\},$$
$$\mathcal{I}_{31} = \{j \in \mathcal{I} : \mathbf{u}_j = 0, j \geq k_2^*\}, \qquad \mathcal{I}_{32} = \{j \in \mathcal{I} : \mathbf{u}_j = q, j \geq k_2^*\}.$$

Denote the overall division by $k_1^*$ and $k_2^*$ by $\mathcal{I}_1 = \mathcal{I}_{11} \cup \mathcal{I}_{12}$, $\mathcal{I}_2 = \mathcal{I}_{21} \cup \mathcal{I}_{22}$, and $\mathcal{I}_3 = \mathcal{I}_{31} \cup \mathcal{I}_{32}$.

We claim that for any such $\mathbf{u}$, there exists a vector $\tilde{\mathbf{u}} \in \Gamma(\mathbf{v}) \cup \Gamma(-\mathbf{v})$ such that $\sin \theta(\mathbf{v}, \mathbf{u}) \geq \sin \theta(\mathbf{v}, \tilde{\mathbf{u}})$, which is sufficient to complete the proof since $\mathbf{u}^*$ is the minimizer of $\sin \theta(\mathbf{v}, \mathbf{u})$ for all $\mathbf{u} \in \Gamma(\mathbf{v}) \cup \Gamma(-\mathbf{v})$. We will show how to find such vector $\tilde{\mathbf{u}}$ by examining the following two cases:

(Case 1) $\langle \mathbf{v}, \mathbf{u} \rangle \geq 0$:

Let $c_1 = |\mathcal{I}_{21}| - |\mathcal{I}_1|$ and $c_2 = |\mathcal{I}_{22}| - |\mathcal{I}_3|$. The claim is proved by setting $\tilde{\mathbf{u}} = \sigma_{t_{k_1^*+c_1}, t_{k_2^*-c_2}}(\mathbf{v})$, which is justified by the following two observations.

First, observe that $\|\tilde{\mathbf{u}}\|_2 \leq \|\mathbf{u}\|_2$ because $\|\tilde{\mathbf{u}}\|_2^2 + |\mathcal{I}_{12}| + q^2 |\mathcal{I}_{32}| = \|\mathbf{u}\|_2^2$.

Second, write $\langle \mathbf{v}, \mathbf{u} \rangle$ as

$$\langle \mathbf{v}, \mathbf{u} \rangle = \langle \mathbf{v}, \mathbf{u}^* \rangle + q \left( -\sum_{j \in \mathcal{I}_1} \mathbf{v}_j + \sum_{j \in \mathcal{I}_{21} \cup \mathcal{I}_{32}} \mathbf{v}_j \right) + \left( \sum_{j \in \mathcal{I}_3} \mathbf{v}_j + \sum_{j \in \mathcal{I}_{12} \cup \mathcal{I}_{22}} (-\mathbf{v}_j) \right). \quad (1)$$

Notice that some terms of the summation in Equation (1) are negative, in particular,

$$\sum_{j \in \mathcal{I}_{12}} (-\mathbf{v}_j) + \sum_{j \in \mathcal{I}_{32}} q\mathbf{v}_j < 0,$$

since $\mathbf{v}_j > 0$ for all $j \in \mathcal{I}_{12}$, and $\mathbf{v}_j < 0$ for all $j \in \mathcal{I}_{32}$.

Therefore, we have

$$\langle \mathbf{v}, \mathbf{u} \rangle \leq \langle \mathbf{v}, \mathbf{u}^* \rangle + q \left( -\sum_{j \in \mathcal{I}_1} \mathbf{v}_j + \sum_{j \in \mathcal{I}_{21}} \mathbf{v}_j \right) + \left( \sum_{j \in \mathcal{I}_3} \mathbf{v}_j + \sum_{j \in \mathcal{I}_{22}} (-\mathbf{v}_j) \right). \qquad (2)$$

Since $\mathbf{v}$ is sorted non-increasingly, the latter two terms in (2) are smaller than

$$q \left( -\sum_{j=1}^{|\mathcal{I}_1|} \mathbf{v}_{k_1^*-j} + \sum_{j=1}^{|\mathcal{I}_{21}|} \mathbf{v}_{k_1^*-|\mathcal{I}_1|+j} \right) + \left( \sum_{j=1}^{|\mathcal{I}_3|} \mathbf{v}_{k_2^*+j} + \sum_{j=1}^{|\mathcal{I}_{22}|} (-\mathbf{v}_{k_2^*+|\mathcal{I}_3|-j}) \right).$$

That is,

$$\langle \mathbf{v}, \mathbf{u} \rangle \leq \langle \mathbf{v}, \mathbf{u}^* \rangle + q \left( -\sum_{j=1}^{|\mathcal{I}_1|} \mathbf{v}_{k_1^*-j} + \sum_{j=1}^{|\mathcal{I}_{21}|} \mathbf{v}_{k_1^*-|\mathcal{I}_1|+j} \right) + \left( \sum_{j=1}^{|\mathcal{I}_3|} \mathbf{v}_{k_2^*+j} + \sum_{j=1}^{|\mathcal{I}_{22}|} (-\mathbf{v}_{k_2^*+|\mathcal{I}_3|-j}) \right)$$

$$= \langle \mathbf{v}, \tilde{\mathbf{u}} \rangle$$

Hence, we have $0 \leq \cos\theta(\mathbf{v}, \mathbf{u}) \leq \cos\theta(\mathbf{v}, \tilde{\mathbf{u}})$, which is equivalent to $\sin\theta(\mathbf{v}, \mathbf{u}) \geq \sin\theta(\mathbf{v}, \tilde{\mathbf{u}})$ due to the non-negativity of $\sin\theta(\cdot, \cdot)$.

(Case 2) $\langle \mathbf{v}, \mathbf{u} \rangle < 0$:

Let $c_1 = |\{j \in \mathcal{I}_{21} : \mathbf{v}_j < 0\}| + |\mathcal{I}_{32}|$ and $c_2 = |\{j \in \mathcal{I}_{22} : \mathbf{v}_j > 0\}| + |\mathcal{I}_{12}|$. The claim is proved by setting $\tilde{\mathbf{u}} = \sigma_{t'_{c_1}, t'_{c_2}}(-\mathbf{v})$, which is justified in the below two observations.

First, observe that $\|\tilde{\mathbf{u}}\|_2 \leq \|\mathbf{u}\|_2$ because

$$\|\tilde{\mathbf{u}}\|_2^2 + q^2|\{j \in \mathcal{I}_{21} : \mathbf{v}_j \geq 0\}| + |\{j \in \mathcal{I}_{22} : \mathbf{v}_j \leq 0\}| = \|\mathbf{u}\|_2^2.$$

Second, write $\langle \mathbf{v}, \mathbf{u} \rangle$ by Equation (1) as

$$\langle \mathbf{v}, \mathbf{u} \rangle = \langle \mathbf{v}, \mathbf{u}^* \rangle + q \left( -\sum_{j \in \mathcal{I}_1} \mathbf{v}_j + \sum_{j \in \mathcal{I}_{21} \cup \mathcal{I}_{32}} \mathbf{v}_j \right) + \left( \sum_{j \in \mathcal{I}_3} \mathbf{v}_j + \sum_{j \in \mathcal{I}_{12} \cup \mathcal{I}_{22}} (-\mathbf{v}_j) \right). \qquad (3)$$

Notice that some terms of the summation in Equation (3) are non-negative, in particular

$$\sum_{j \in \mathcal{I}_{21}, \mathbf{v}_j \geq 0} q\mathbf{v}_j + \sum_{j \in \mathcal{I}_{22}, \mathbf{v}_j \leq 0} (-\mathbf{v}_j) \geq 0.$$

Therefore, by letting $\mathcal{I}_{21}^- = \{i \in \mathcal{I}_{21}, \mathbf{v}_i < 0\}$ and $\mathcal{I}_{22}^+ = \{i \in \mathcal{I}_{22}, \mathbf{v}_i > 0\}$, we have

$$\langle \mathbf{v}, \mathbf{u} \rangle \geq \langle \mathbf{v}, \mathbf{u}^* \rangle + q \left( -\sum_{j \in \mathcal{I}_1} \mathbf{v}_j + \sum_{j \in \mathcal{I}_{32} \cup \mathcal{I}_{21}^-} \mathbf{v}_j \right) + \left( \sum_{j \in \mathcal{I}_3} \mathbf{v}_j - \sum_{j \in \mathcal{I}_{12} \cup \mathcal{I}_{22}^+} \mathbf{v}_j \right) \qquad (4)$$

$$\geq q \sum_{j \in \mathcal{I}_{32} \cup \mathcal{I}_{21}^-} \mathbf{v}_j - \sum_{j \in \mathcal{I}_{12} \cup \mathcal{I}_{22}^+} \mathbf{v}_j \qquad (5)$$

$$\geq - \left( q \sum_{j=1}^{|\mathcal{I}_{32} \cup \mathcal{I}_{21}^-|} t'_j - \sum_{j=|\mathcal{I}_{12} \cup \mathcal{I}_{22}^+|+1}^{n} t'_j \right) = -\langle \mathbf{v}, \tilde{\mathbf{u}} \rangle,$$

where Inequalities (4) and (5) hold because $\mathcal{I}_1 \subseteq [k_1^*]$ and $\mathcal{I}_3 \subseteq [k_2^*, \cdots, n]$.

Hence, we have $0 \geq \cos\theta(\mathbf{v}, \mathbf{u}) \geq \cos\theta(\mathbf{v}, \tilde{\mathbf{u}})$, which is equivalent to $\sin\theta(\mathbf{v}, \mathbf{u}) \geq \sin\theta(\mathbf{v}, \tilde{\mathbf{u}})$ due to the non-negativity of $\sin\theta(\cdot, \cdot)$. $\qquad \square$

# B Proof of Theorem 1

**Proof:** Let $\mathbf{u}$ be the random variable defined in Section 6.2, such that $\mathbf{u}_i \sim q\,\mathrm{Bernoulli}(|\mathbf{v}_i|/q)$ for positive coordinates $\mathbf{v}_i > 0$, $\mathbf{u}_i \sim (-1)\mathrm{Bernoulli}(|\mathbf{v}_i|)$ for negative coordinates $\mathbf{v}_i < 0$, and $\mathbf{u}_i = 0$ if $\mathbf{v}_i = 0$. For convenience, we define

$$g(x) = \begin{cases} q, & x > 0 \\ -1, & x < 0 \\ 0, & x = 0. \end{cases}$$

We are interested in analyzing the expectation of $\mathbf{u}^T A \mathbf{u}/\mathbf{u}^T \mathbf{u}$, which is given by

$$\mathbb{E}\left[\frac{\mathbf{u}^T A \mathbf{u}}{\mathbf{u}^T \mathbf{u}}\right] = \sum_{(k_1,k_2):1\leq k_1+k_2\leq n} \mathbb{E}\left[\frac{\mathbf{u}^T A \mathbf{u}}{\mathbf{u}^T \mathbf{u}} \mid \mathbf{u}^T \mathbf{u} = qk_1 + k_2\right]\mathbb{P}(\mathbf{u}^T \mathbf{u} = qk_1 + k_2)$$

$$= \sum_{(k_1,k_2):1\leq k_1+k_2\leq n} \frac{\mathbb{E}\left[\mathbf{u}^T A \mathbf{u} \mid \mathbf{u}^T \mathbf{u} = qk_1 + k_2\right]\mathbb{P}(\mathbf{u}^T \mathbf{u} = qk_1 + k_2)}{qk_1 + k_2}. \quad (6)$$

The term $\mathbb{E}\left[\mathbf{u}^T A \mathbf{u} \mid \mathbf{u}^T \mathbf{u} = qk_1 + k_2\right]\mathbb{P}(\mathbf{u}^T \mathbf{u} = qk_1 + k_2)$ in Equation (6), can be written as

$$\sum_{i\neq j} A_{i,j}g(\mathbf{v}_i)g(\mathbf{v}_j)\mathbb{P}(\mathbf{u}_i = g(\mathbf{v}_i), \mathbf{u}_j = g(\mathbf{v}_j) \mid \mathbf{u}^T \mathbf{u} = qk_1 + k_2)\mathbb{P}(\mathbf{u}^T \mathbf{u} = qk_1 + k_2), \quad (7)$$

and using Bayes' theorem we can re-write Equation (7) as

$$\sum_{i\neq j} A_{i,j}g(\mathbf{v}_i)g(\mathbf{v}_j)\mathbb{P}(\mathbf{u}^T \mathbf{u} = qk_1+k_2 \mid \mathbf{u}_i = g(\mathbf{v}_i), \mathbf{u}_j = g(\mathbf{v}_j))\mathbb{P}(\mathbf{u}_i = g(\mathbf{v}_i), \mathbf{u}_j = g(\mathbf{v}_j)). \quad (8)$$

By Equations (6) and (8) and since $g(\mathbf{v}_i)g(\mathbf{v}_j)\mathbb{P}(\mathbf{u}_i = g(\mathbf{v}_i), \mathbf{u}_j = g(\mathbf{v}_j)) = \mathbf{v}_i\mathbf{v}_j$, we have

$$\sum_{(k_1,k_2):1\leq k_1+k_2\leq n} \frac{\sum_{i\neq j} A_{i,j}\mathbf{v}_i\mathbf{v}_j\mathbb{P}(\mathbf{u}^T \mathbf{u} = qk_1 + k_2 \mid \mathbf{u}_i = g(\mathbf{v}_i), \mathbf{u}_j = g(\mathbf{v}_j))}{qk_1 + k_2}$$

$$= \sum_{i\neq j} A_{i,j}\mathbf{v}_i\mathbf{v}_j \sum_{(k_1,k_2):1\leq k_1+k_2\leq n} \frac{\mathbb{P}(\mathbf{u}^T \mathbf{u} = qk_1 + k_2 \mid \mathbf{u}_i = g(\mathbf{v}_i), \mathbf{u}_j = g(\mathbf{v}_j))}{qk_1 + k_2}$$

$$= \sum_{i\neq j} A_{i,j}\mathbf{v}_i\mathbf{v}_j\mathbb{E}\left[\frac{1}{\mathbf{u}^T \mathbf{u}} \mid \mathbf{u}_i = g(\mathbf{v}_i), \mathbf{u}_j = g(\mathbf{v}_j)\right]. \quad (9)$$

As the reciprocal function is convex, we apply Jensen's inequality to Equation (9) to obtain

$$\mathbb{E}\left[\frac{\mathbf{u}^T A \mathbf{u}}{\mathbf{u}^T \mathbf{u}}\right] \geq \frac{\sum_{i\neq j} A_{i,j}\mathbf{v}_i\mathbf{v}_j}{\mathbb{E}\left[\mathbf{u}^T \mathbf{u} \mid \mathbf{u}_i = g(\mathbf{v}_i), \mathbf{u}_j = g(\mathbf{v}_j)\right]}. \quad (10)$$

To estimate the denominator in Equation (10), we compute

$$\mathbb{E}\left[\mathbf{u}^T \mathbf{u} \mid \mathbf{u}_i = g(\mathbf{v}_i), \mathbf{u}_j = g(\mathbf{v}_j)\right] = g(\mathbf{v}_i)^2 + g(\mathbf{v}_j)^2 + \sum_{k\neq i, k\neq j} g(\mathbf{v}_k)^2 \cdot \frac{|\mathbf{v}_k|}{|g(\mathbf{v}_k)|}$$

$$\leq \max\left(q\sqrt{n-2}, 2q^2 + q\frac{n-2}{\sqrt{n}}\right). \quad (11)$$

Combining (10) and (12) we get

$$\mathbb{E}\left[\frac{\mathbf{u}^T A \mathbf{u}}{\mathbf{u}^T \mathbf{u}}\right] \geq \frac{\sum_{i\neq j} A_{i,j}\mathbf{v}_i\mathbf{v}_j}{\max\left(q\sqrt{n-2}, 2q^2 + q\frac{n-2}{\sqrt{n}}\right)} = \frac{\lambda_1(A)}{\max\left(q\sqrt{n-2}, 2q^2 + q\frac{n-2}{\sqrt{n}}\right)}. \quad (12)$$

Hence, the expected approximation ratio is

$$\mathcal{O}(q\sqrt{n})\mathbb{E}\left[\frac{\mathbf{u}^T A \mathbf{u}}{\mathbf{u}^T \mathbf{u}}\right] \geq \lambda_1(A) \geq OPT,$$

where $OPT$ is the optimum of MAX-DRQ. $\qquad\square$

## C Proof of Lemma 2

**Proof:** Consider a graph $G = (V, E)$ consisting of $|V| = n = 2c + 1$ nodes, for some $c \geq 1$, where $2c$ nodes form a negative clique and the extra node $v$ is negatively connected to $c$ of the nodes in the clique. Let $A$ be the signed adjacency matrix of $G$. We will show the problem instance defined on $G$ results in an optimal value of MAX-DRQ equal to $OPT = \mathcal{O}(1)$, while $\lambda_1(A)$ is $\Omega(\sqrt{n})$.

Any solution $\mathbf{u} \in \{0, -1, q\}^n$ to MAX-DRQ defines the two sets $S_p = \{i : \mathbf{u}_i = q\}$ and $S_n = \{i : \mathbf{u}_i = -1\}$. We claim that $\max_{\mathbf{u} \in \{0, -1, q\}^n} \mathbf{u}^T A \mathbf{u} / \mathbf{u}^T \mathbf{u} \leq 2$, and will show it by considering 3 cases:

(Case 1) $v \notin S_p \cup S_n$:

$$\frac{\mathbf{u}^T A \mathbf{u}}{\mathbf{u}^T \mathbf{u}} = \frac{-q^2 \overbrace{|S_p|(|S_p| - 1)}^{2|E(S_p)|} + q \overbrace{2|S_p||S_n|}^{2|E(S_p, S_n)|} - \overbrace{|S_n|(|S_n| - 1)}^{2|E(S_n)|}}{q^2|S_p| + |S_n|}$$
$$= \frac{-(q|S_p| - |S_n|)^2 + q^2|S_p| + |S_n|}{q^2|S_p| + |S_n|}. \tag{13}$$

Let $r = q|S_p| - |S_n|$ and let $\epsilon = r/q|S_p| \leq 1$. Then, Equation (13) can be written as

$$\frac{\mathbf{u}^T A \mathbf{u}}{\mathbf{u}^T \mathbf{u}} = \frac{q(q+1)|S_p| - r(r+1)}{|S_p|q(q+1) - r}$$
$$= \frac{(q+1) + \frac{1}{4(q|S_p|)}}{(q+1) - \epsilon} - \frac{(r + \frac{1}{2})^2}{q|S_p|(q + 1 - \epsilon)}$$
$$\leq \frac{(q+1) + \frac{1}{4(q|S_p|)}}{(q+1) - \epsilon}$$
$$\leq \frac{q + 2}{q} \leq 2 = \mathcal{O}(1).$$

(Case 2) $v \in S_p$:

$$\mathbf{u}^T A \mathbf{u} = -q^2 \left( \underbrace{(|S_p| - 1)(|S_p| - 2)}_{2|E(S_p \setminus \{v\})|} + 2|E(\{v\}, S_p)| \right)$$
$$+ q \left( \underbrace{2(|S_p| - 1)|S_n|}_{2|E(S_p \setminus \{v\}, S_n)|} + 2|E(\{v\}, S_n)| \right) - \underbrace{|S_n|(|S_n| - 1)}_{2|E(S_n)|}$$
$$= -(q(|S_p| - 1) - |S_n|)^2 + |S_n|$$
$$\quad + q^2(|S_p| - 1) + 2q|E(\{v\}, S_n)| - 2q^2|E(\{v\}, S_p)|$$
$$\leq -(q(|S_p| - 1) - |S_n|)^2 + |S_n| + q^2(|S_p| - 1) + 2q|S_n|. \tag{14}$$

Let $r = q(|S_p| - 1) - |S_n|$ and write Equation (14) as

$$\mathbf{u}^T A \mathbf{u} = -(r - q)^2 + q(q + 3)(|S_p| - 1) + q^2 - r$$
$$\leq q(q + 3)(|S_p| - 1) + q^2 - r. \tag{15}$$

By (15) and letting $\epsilon = r/q(|S_p| - 1) \leq 1$, we have

$$\frac{\mathbf{u}^T A \mathbf{u}}{\mathbf{u}^T \mathbf{u}} \leq \frac{q(q + 3)(|S_p| - 1) + q^2 - r}{q(q + 1)(|S_p| - 1) + (q^2 - r)}$$
$$= 1 + \frac{2}{(q + 1) + (q/(|S_p| - 1) - \epsilon)} \leq 2 = \mathcal{O}(1).$$

(Case 3) $v \in S_n$:

$$\mathbf{u}^T A \mathbf{u} = q^2 |S_p|(|S_p| - 1) + q \left( \underbrace{|S_p|(|S_n| - 1)}_{2|E(S_n \setminus \{v\}, S_p)|} + 2|E(\{v\}, S_p)| \right)$$

$$- \left( \underbrace{(|S_n| - 1)(|S_n| - 2)}_{2|E(S_n \setminus \{v\})|} + 2|E(\{v\}, S_n)| \right)$$

$$= -\left( q|S_p| - (|S_n| - 1) \right)^2 + q^2 |S_p| + |S_n| - 1$$
$$+ 2q|E(\{v\}, S_p)| - 2|E(\{v\}, S_n)|$$
$$\leq -\left( q|S_p| - (|S_n| - 1) \right)^2 + q^2 |S_p| + |S_n| - 1 + 2q|S_p|. \qquad (16)$$

Let $r = q|S_p| - (|S_n| - 1)$ and write Inequality (16) as

$$\mathbf{u}^T A \mathbf{u} = -(r + \frac{1}{2})^2 + q(q + 3)|S_p| + \frac{1}{4}$$

$$\leq q(q + 3)|S_p| + \frac{1}{4}. \qquad (17)$$

By (17) and letting $\epsilon = (r + 1)/q|S_p| \leq 1$, we have

$$\frac{\mathbf{u}^T A \mathbf{u}}{\mathbf{u}^T \mathbf{u}} \leq \frac{q(q + 3)|S_p| + \frac{1}{4}}{q(q + 1)|S_p| - (r + 1)}$$

$$= 1 + \frac{2 + 1/(4q|S_p|) + \epsilon}{(q + 1) - \epsilon}$$

$$\leq 2 = \mathcal{O}(1).$$

Therefore, we know that the optimal solution $OPT$ of MAX-DRQ is $\mathcal{O}(1)$. However, consider a vector $\mathbf{x} \in \mathbb{R}^n$ such that

$$\mathbf{x} = \left[ \sqrt{\frac{n + 1}{2n}}, \underbrace{\frac{1}{\sqrt{2n}}, \cdots, \frac{1}{\sqrt{2n}}}_{c \text{ entries}}, \underbrace{\frac{-1}{\sqrt{2n}}, \cdots, \frac{-1}{\sqrt{2n}}}_{c \text{ entries}} \right], \qquad (18)$$

where the first entry of $\mathbf{x}$ corresponds to $v$. Then, the vector $\mathbf{x}$ defined in Equation (18) gives

$$\frac{\mathbf{x}^T A \mathbf{x}}{\mathbf{x}^T \mathbf{x}} = \frac{\sqrt{n + 1}(n - 1)}{2n} + \frac{n - 1}{2n}$$

$$= \frac{\sqrt{n + 1} + 1}{2} - \frac{\sqrt{n + 1} + 1}{2n}$$

$$= \Omega(\sqrt{n}).$$

As $\lambda_1(A) = \max_{\mathbf{x} \in \mathbb{R}^n \setminus \{0\}} \mathbf{x}^T A \mathbf{x}/\mathbf{x}^T \mathbf{x}$, we have shown $\lambda_1(A) \geq OPT \cdot \Omega(\sqrt{n})$. $\qquad \square$

# D   Experiment Results

## D.1   Dataset

`WoW-EP8` [1] is the interaction network of authors in the 8th legislature of the EU Parliament, where edge signs indicate if two authors are collaborative or competitive to each other. `Bitcoin` [4] is the trust-distrust network of users trading on the Bitcoin OTC platform. `WikiVot` [4] collects the positive and negative votes for electing Wikipedia admins. `Referendum` [3] collects the tweets about the Italian constitutional referendum in 2016, and edge signs indicate if two users are classified to have the same stance or not. `Slashdot` [4] is a friend-foe network collected from the Slashdot Zoo feature. `WikiCon` [2] collects the positive and negative iterations of users editing the English Wikipedia. `Epinions` [4] is the trust-distrust network of users on the online social network Epinions. `WikiPol` [5] is the interaction network of users who have edited the English Wikipedia pages about politics.

## D.2   Execution Time

Table 2:  Running times for the results shown in Table 1. All times are shown in seconds. Dashes indicate that a method cannot finish execution due to memory limit exceeded.

| | | WoW-EP8 | Bitcoin | WikiVot | Referendum | Slashdot | WikiCon | Epinions | WikiPol |
|---|---|---|---|---|---|---|---|---|---|
| | $|V|$ | 790 | 5 881 | 7 115 | 10 884 | 82 140 | 116 717 | 131 580 | 138 587 |
| | $|E|$ | 116 009 | 21 492 | 100 693 | 251 406 | 500 481 | 2 026 646 | 711 210 | 715 883 |
| | $|E_-|/|E|$ | 0.2 | 0.2 | 0.2 | 0.1 | 0.2 | 0.6 | 0.2 | 0.1 |
| $k=2$ | SCG-MA | 2 | 1 | 2 | 4 | 10 | 217 | 109 | 25 |
| | SCG-MO | 2 | 1 | 2 | 4 | 11 | 70 | 94 | 15 |
| | SCG-B | 13 | 9 | 21 | 44 | 693 | 3 584 | 1 906 | 1 624 |
| | SCG-R | 4 | 3 | 6 | 17 | 70 | 485 | 37 | 217 |
| | KOCG | 3 | 11 | 16 | 25 | 1 243 | 3 269 | 3 208 | 3 506 |
| | BNC-$k$ | 2 | 1 | 2 | 4 | — | — | — | — |
| | BNC-$(k+1)$ | 2 | 1 | 2 | 4 | — | — | — | — |
| | SPONGE-$k$ | 2 | 5 | 3 | 4 | — | — | — | — |
| | SPONGE-$(k+1)$ | 2 | 11 | 4 | 9 | — | — | — | — |
| $k=6$ | SCG-MA | 3 | 1 | 6 | 16 | 75 | 394 | 132 | 136 |
| | SCG-MO | 3 | 1 | 6 | 18 | 74 | 229 | 107 | 139 |
| | SCG-B | 17 | 29 | 78 | 201 | 3 280 | 10 637 | 5 455 | 5 714 |
| | SCG-R | 3 | 5 | 9 | 21 | 118 | 415 | 219 | 892 |
| | KOCG | 1 | 5 | 8 | 14 | 690 | 1 837 | 1 845 | 1 724 |
| | BNC-$k$ | 2 | 1 | 2 | 4 | — | — | — | — |
| | BNC-$(k+1)$ | 2 | 1 | 2 | 4 | — | — | — | — |
| | SPONGE-$k$ | 2 | 7 | 6 | 20 | — | — | — | — |
| | SPONGE-$(k+1)$ | 2 | 5 | 4 | 26 | — | — | — | — |

## D.3   Detected Group Sizes

Figure 2, extracted from the `Referendum` dataset, shows the typical distribution of the group sizes for all the comparison methods. This pattern is similar to all other datasets except `WoW-EP8`. That is, SCG-MA, SCG-MO, and SCG-R return the largest groups while KOCG-top-1, BNC-$(k+1)$, and SPONGE-$(k+1)$ return the smallest groups.

Figure 2:  The typical group size distribution on all the datasets except `WoW-EP8` when $k=6$.

On the other hand, `WoW-EP8` shows a different group-size distribution, which is shown in Figure 3. All SCG methods and BNC-$k$ find one giant group. By checking the polarity (Table 1 in main paper), their scores are high, so this probably suggests there exists a giant conflicting group in the network.

Figure 3: The group size distribution on `WoW-EP8` when $k = 6$.

### D.4  Deciding $k$

We present a heuristic similar to Elbow Method [6] to decide $k$, which consists of the following steps:

1. Run `SCG` multiple times with different $k$.
2. Draw a `DRQ-Plot`, where the Discrete Rayleigh Quotient (DRQ) values in each run are sorted, and then plot the $i$-th largest DRQ value at the $i$-th location.
3. Decide $k$ to be one of the "knees" of the curve.

The reason why the heuristic works is that, if there exist conflicting groups and the noise-level is not too high, then the leading eigenvector should be indicative of the true conflicting groups and have large DRQ values in the first $k-1$ iterations, while the leading eigenvector only captures noise structures and has low DRQ value after the $k$-th iteration. Therefore, it is expected to see knees of the curve at the $(k-1)$-th iteration.

First, we evaluate the heuristic using m-SSBM under the same setting ($k = 6$, $\ell = 100$, and $n = 2\,000$) by varying $\eta = 0 : 0.1 : 0.6$. The result of detecting the conflicting groups by SCG-MA is depicted in Figure 4. As expected, the most prominent knee is at the 5-th iteration when the noise-level is not too high ($\eta \leq 0.3$). As the noise-level increases ($\eta \geq 0.4$), the knee at the 5-th iteration becomes less obvious and some artificial knees that fit the random noise emerge.

Figure 4: Run SCG-MA with different $k$ on networks generated by m-SSBM ($k = 6$, $\ell = 100$, and $n = 2\,000$). Each setting is repeated 20 times and reported the average.

Finally, we use the heuristic on the real-world datasets to decide $k$ and show the result in Figure 5. Our analysis suggests that `Referendum` has 4 conflicting groups, because the most prominent knee

appears at the 3-th iteration, while on `Epinions`, there are two prominent knees at the 3rd and the 4-th iterations, so there are probably 4 or 5 conflicting groups in the network.

Figure 5: Run SCG-MA Real-world networks with different $k$.