[Reviews · NeurIPS 2020]

Review 1

Summary and Contributions: The authors describe a non-convex optimization problem on signed networks for finding k conflicting subsets: groups of nodes with high positive internal connectivity and negative external connectivity to the other subsets. The objective is cast in terms of the Laplacian on a complete graph, motivating a spectral optimization algorithm. The spectral algorithm is coupled to two rounding approaches. The resulting algorithm performs better than competing methods in terms of both the optimization metric and the recovery of ground-truth subgroups.

Strengths: The spectral formulation of the problem is nice, and the rounding algorithms are both novel and highly effective. The first rounding algorithm (minimum angle heuristic) shows the strongest performance in several trials, but the randomized rounding algorithm comes with an expected tightness gap. The experiments based on the new approach are impressive both in terms of their effectiveness and scalability -- the previous approaches ran out of RAM on several of the test problems.

Weaknesses: As the authors note, the randomized algorithm fares significantly worse than the minimum angle heuristic, despite the theoretical guarantees. This is not a sign that either the theory or the implementation is wrong, only that there is work yet to be done.

Correctness: Yes, they appear to be.

Clarity: Yes.

Relation to Prior Work: Yes.

Reproducibility: Yes

Additional Feedback: --- Added after rebuttal --- I have read the other reviews and the author rebuttal, and remain enthusiastic about this work. I have not changed my score.


Review 2

Summary and Contributions: This works seeks to focus on the problem of detecting dense conflicting groups within a graph containing both positive and negative links (i.e., signed graph). More specifically, they focus on extending the work in [4] by generalizing to allowing a variable number (i.e., k) groups as compared to just 2 in the prior work. As compared to more traditional community detection in signed networks where the authors highlight correlation clustering, signed Laplacian methods, and k-way balanced normalized cuts for clustering the signed graphs. The novelty in this work is the allowance of nodes to be neutral rather than forcing all nodes to join one of the k polarized groups. Based on this new problem they convert the problem to instead optimizing for the maximum discrete Rayleigh quotient (i.e., MAX-DRQ). Then, solutions are presented for the converted problem one of which is a randomized algorithm. Empirical results are made with existing methods although, due to the new problem setting, they select appropriate baselines to compare with. Synthetic experiments are done using the signed stochastic block model m-SSBM presented previously in [4].

Strengths: * Detecting conflicting groups in online social networks is an interesting and important topic. * The solutions to the MAX-DRQ are novel and well presented, especially the given deterministic rounding algorithm SCG-MA. * The authors have performed experiments on many datasets.

Weaknesses: * The decision to use the weighting of k-1 in Eq (1) does not seem to be well motivated. The argument in regards to the assumption of intra-group and inter-group having the same edge density seems not something that is found in reality and not empirically supported. [Note: The authors have addressed this concern.] * Does this work scale to popular online social media? The experiments were performed on real-world signed networks on a rather small to medium scale (e.g., 1k to 100k), but prior works such as ``Finding large balanced subgraphs in signed networks’’ (WWW2020) tested their scalability up to 34 million edges. It is likely that all but SCG-B can scale, but would be good to include. [Note: The authors noted some running time analysis with most in the hours order of magnitude (except SCG-B as expected taking longer than 1 day without finishing), but mention leaving this as future work and will not include in this paper.] * Are there any datasets that can be performed with ground truth? The real-world are utilizing the polarity objective defined in Equation 6, but the baselines are not designed to optimize this specifically. For example, in Figure 1, we see using this polarity score (right) the prior works (e.g., SPONGE) are significantly worse than the proposed methods, but when considering the synthetic ground truth and F1 (left) we see that SPONGE performs quite comparable (slightly above or below across values of eta (which controls the edge probabilities). Perhaps there can be some political datasets, e.g., congressional datasets, that could be used here? [This is still a concern and hopefully can be addressed in their future work (if possible).]

Correctness: The methods and empirical settings are correct.

Clarity: The paper is well motivated for why the problem is both interesting and different compared to existing works. Overall it is well written.

Relation to Prior Work: This work attempts to make it clear how their work differs from the closest work they extend, namely [4], by removing the limiting factor of k=2 to a more generalized setting.

Reproducibility: Yes

Additional Feedback: There seems to be an unbalance in some of the referenced prior works. For example, there are 3 papers on fake news, while there would be numerous more recent related works in community detection and node clustering on signed graphs, such as ``Attributed Signed Network Embedding” (CIKM2017) that use network embedding to discover communities in signed graphs, and not mentioned. Please note that some works in the references that appear to not be referenced in the actual paper, e.g., ``Stance Evolution and Twitter Interactions in an Italian Political Debate." Thank you for your response. I have updated the review accordingly.


Review 3

Summary and Contributions: The paper presents an approach for identifying k groups of nodes that resemble some sort of balance according to structural balance theory. This work stands out, because it is not aiming at identifying a partition of the set of nodes, but at identifying k non-overlapping sets of nodes where most of positive edges are inside the sets, and most of the edges are between the sets. The paper is mainly inspired by contributions from the correlation-clustering community, and proposes a numerical scheme that is shown empirically to be faster than clustering approaches. Further, it shows that the objective functions and the corresponding continuous relaxations of other approaches stand apart from the one proposed by the authors.

Strengths: An interesting and relevant task is considered in this paper, mainly the identification of k groups of nodes, where most of the positive edges are inside the sets, and most of negative edges are between the sets. This is inspired by the notion of structural social balance, which is the standard notion followed by signed graph based approaches. Furthermore, the authors are inspired by works on correlation clustering, which contrary to signed graph clustering approaches, do not require to set the number of clusters to be defined a priori, i.e. the number of clusters is automatically estimated via suitable objective function. The authors observe that this approach is initially NP-hard, and propose a computable approach that indeed is shown to be faster via numerical experiments. Further, a quick analysis on the number of groups to identify is presented in the supplementary material.

Weaknesses: It would be interesting to observe to what extend the identified groups do match certain ground truth. For instance, since the authors are considering a modified Stochastic Block model, it would relevant to see under what settings the proposed approach is able to identify the right solution. At the moment the only evaluation available is in terms of the objective function, leaving unclear if this is sufficiently close to the ground truth solution. For instance, what is the value of the objective function on the ground-truth solution, and hence, how close are is the output of the propose approach to this value? The paper provides a theoretical analysis on the approximation to the ground truth solution, to later provide numerical experiments under the m-SSBM. It would be interesting to provide a theoretical analysis showing under which parameters of the m-SSBM the right solution is identified/approximated. Traditionally, an analysis in expectation is of interest in the field of graph-based clustering. Is is then possible to do a similar analysis with the proposed approach in this paper? See for instance Theorems 1,2,3 in [12] or Theorem5,6 in [30]. Further, an analysis in sparsity would provide insights into how well the proposed approach behaves. It is known that most of graph methods present problems for very sparse graphs. Hence, approaches based on the Bethe Hessian or Non-Back Tracking Operators are often of high interest in this regime. One method skipped by the authors is indeed based on such operators (https://arxiv.org/pdf/1502.00163.pdf). It would be highly interesting to see how the Bethe Hessian performs in comparison to the propose approach.

Correctness: The numerical evaluations seem to be methodologically correct. Yet, it would be interesting if error bars could be provided. It is clear that the proposed approach presents a performance that stands apart from the rest, yet error bars would provide a notion on the stability of the proposed approach. Since the authors observed that certain clustering approaches fail by returning a large component as a cluster/group, it would be interesting to know why the proposed approach does not fall in to this issue. Is there any particular reason why this does not happen?

Clarity: The paper is well written and it is easy to follow.

Relation to Prior Work: The paper presents a nice overview of methods that are related to the task that they study. Considering the task of clustering in signed networks as a reference provides a good notion on how the proposed task is rather different, and how numerically both problems identify different solutions. I would suggest to consider the approach proposed in https://arxiv.org/pdf/1502.00163.pdf, as it could provide even further advantages to the proposed approach.

Reproducibility: Yes

Additional Feedback:


Review 4

Summary and Contributions: The main problem the author solved in this article is to detect k dense conflicting -group which ask the edges are mostly positive in each group, whereas the edges are mostly negative with other k-1 groups, and allow a subset of nodes to be neutral with respect to the conflict structure in a signed networks. The author extend special case (k=2) to general case (arbitrary values of k), and based on the spectral properties of Laplacian of a complete graph to derive a novel optimization framework called SCG, next, the author successfully converted the problem detecting each k conflicting groups to solve the maximum discrete Rayleigh’s quotient (MAX-DRQ) problem. Furthermore, the authors presented two spectral methods to find approximate solutions to MAX-DRQ problem, one is a deterministic algorithm and the other is a randomized algorithm. In results on real-world networks, their methods i.e. SCG-MA and SCG-MO effectively find solutions of higher quality, faster speed and lower memory demand compared to other strong baselines, like KOCG, BNC, SPONGE on two special case k=2 and 6. In results on synthetic graphs, SCG-MA and SCG-MO performed well in recovering ground-truth conflicting groups even compared with competitive SPONGE with respect to recovery rate. *** After Rebuttal*** I have read other reviews and feedback. The authors have addressed my comments well. I increase the score to 7.

Strengths: 1. In this article, the author’s approach for detecting k-conflicting groups in signed networks extends the formulation of Bonchi et al. to arbitrary values of k. 2. The problem is solved by linking to maximum discrete Rayleigh quotient of the group, which had not been linked before. 3. MAX-DRQ is supported by O(n^(1/2))-approximation.

Weaknesses: 1. Can experiments be done to analyze the mean ratio of the number of negative edges in each group and the number of positive edges between groups after search algorithm. 2. In continuation, with this additional suggested analysis it would allow us to see how well the experimental results match our expectation and further test the accuracy of the proposed algorithm. This is the major concern of the work since the current evaluation is based upon a metric defined in this work and thus provides less confidence in the empirical results presented. 3. Some of the formulations were not as easy to follow without referencing the supplemental materials, although this is done for sake of space in the main work perhaps it can be balanced.

Correctness: To my knowledge the the methods are correct and novel. The empirical methods are also correct, but unsure if they are a fair comparison.

Clarity: The paper is written well, but at some points it seems almost necessary to reference the supplemental.

Relation to Prior Work: The prior works seems to be well covered, especially in regards to how this work extends [4] from k=2 to a more general method.

Reproducibility: Yes

Additional Feedback:

[Author Response · NeurIPS 2020]

We thank all the reviewers for their high-quality and constructive feedback! We hope that we address all the concerns below in a satisfactory manner.

**Reply to Reviewer 2.** • Regarding the normalization with $(k-1)$ in Eq. (1), we do not assume that inter-group and intra-group edge densities are equal. Instead, we motivate our normalization by the following reasoning: Suppose that group sizes and inter-group and intra-group edge densities stay fixed (but not necessarily equal) as $k$ increases. Since the number of inter-group edges grows quadratically with $k$ and the number of intra-group edges grows linearly with $k$, if Eq. (1) is unweighted, the objective will be quickly dominated by the number of inter-group edges. We would also like to point out that, unlike clusters in unsigned networks where the intra-group edge density is usually larger than the inter-group edge density, this is not the case for conflicting groups in signed networks. This is indeed the case for the solutions found in all our problem instances, except one (SCG-B with $k = 6$). • Thank you for mentioning scalability. We followed the approach described in the WWW2020 paper to augment WikiCon to 1.1 M nodes and 32.7 M edges, while preserving the ratio of negative edges, and run SCG to detect $k = 6$ conflicting groups on the same machine we reported in our submission. SCG-MA, SCG-MO, SCG-R complete in 1.7 h, 1.1 h, and 2.3 h, respectively, and SCG-B fails to complete in 1 day. We will include additional scalability experiments in the next version of our paper. • Regarding evaluation on datasets with ground truth, unfortunately, we are not aware of such data. • Finally, we will balance our references and consider citing the CIKM2017 paper you pointed out.

**Reply to Reviewer 3.** • Thank you for the suggestion to analyze the recovery condition in m-SSBM and sparse regime, and providing detailed examples and references! We will present this interesting direction in the future-work section. • Q: *Is there any particular reason why the proposed approach does not fail by returning a large component as a cluster/group?* A: We first remind our objective in Eq. (6), which, after ignoring the weighting between inter/intra-group edges, can be expressed as ($\#\{$edges satisfying Property 1$\} - \#\{$edges violating Property 1$\}$) divided by $\#\{$nodes in all conflicting groups$\}$. Intuitively we are seeking for small-size conflicting groups with many "consistent" edges. For simplicity, consider m-SSBM with $\eta = 0$ and let $\{S_j^*\}_{j=1}^k$ be the ground-truth groups. Adding any additional node to a group $S_h^*$ will only decrease the objective score. • Finally, for polarity-score error bars and a comparison to the ground-truth groups, please see below Figures (a) and (b).

**Reply to Reviewer 4.** • Q: *Can experiments be done to analyze the mean ratio of the number of negative edges in each group and the number of positive edges between groups after search algorithm?* A: Assuming that the suggested measure is the following

$$\texttt{MeanDisagreementRatio}(S_1,\cdots,S_k) = \frac{1}{k}\sum_{h=1}^{k}\underbrace{\frac{|\{(i,j)\in E_-\cap E(S_h)\}| + |\{(i,j)\in E_+\cap E(S_h,\cup_{\ell\neq h}S_\ell)\}|}{|E(S_h,\cup_{\ell=1}^{k}S_\ell)|}}_{\texttt{DisagreementRatio of group } S_h},$$

we have computed the measure in our data, and the results are shown below in Figure (c). The error bars show the variance of the `MeanDisagreementRatio` measure on graphs generated by m-SSBM. We observe that KOCG-top-1 has almost always the lowest `MeanDisagreementRatio` score, even lower than the score of the ground-truth solution (in aqua blue). This result suggests that the `MeanDisagreementRatio` measure can be easily hacked by methods like KOCG-top-1, which find solutions of very small size. The reason that `MeanDisagreementRatio` is not a reliable measure, in our opinion, is that it can favor very small/sparse solutions. For example, for $k = 2$, a solution consisting of a single negative edge would be optimal. More generally, groups with purely positive intra-group and/or negative inter-group edges would be optimal for any $k$, regardless of their size or density. On the contrary, our Polarity score cannot be easily hacked by small groups with few edges, since it is unlikely for groups with few edges to have large score. Hence, thank you for this suggestion, but we believe that our proposed objective is preferable. • Regarding some formulations are hard to follow without referencing the Supplementary, we guess you mean Section 6.1. If the paper gets accepted, there will be an additional page and we will expand the discussion and try to make the section self-contained.

(a) F1-Score vs $\eta$     (b) Polarity vs $\eta$     (c) Mean Disagreement Ratio vs $\eta$



[Meta-Review · NeurIPS 2020]

The reviewers were independently happy with the paper. Their reviews raised some concerns and points of confusion that the author response does a good job addressing, so I encourage the authors to incorporate those points into the paper.